# A New Zero-Voltage Zero-Current Switching Converter with Minimum Duty Cycle Loss

Yuting Wang, Yong Shi * and Kexin Xu

School of Electrical and Control Engineering, Shaanxi University of Science and Technology, Xi'an 710021, China; 210611001@sust.edu.cn (Y.W.); 210611011@sust.edu.cn (K.X.)
* Correspondence: shiyong@sust.edu.cn

**Abstract:** Zero-voltage zero-current switching (ZVZCS) phase-shifted full-bridge (PSFB) converters have been widely used in high-power applications because of their high efficiency, low price, and easy control. Currently, the biggest problem with PSFB converters in operation is their high duty cycle loss. With the increase in current, duty cycle loss grows and degrades their performance. Focusing on this problem, a new ZVZCS PSFB converter is proposed in this paper. This topology adds an auxiliary circuit to minimize duty cycle loss. Moreover, the lagging-leg switches can obtain zero-current switching (ZCS) easily with the help of the circuit. The auxiliary circuit is built of four metal-oxide-semiconductor field-effect transistors (MOSFETs) and an auxiliary transformer, and extra voltage can be added to the primary coils when the direct-current (DC) voltage is small. This paper discusses its operation principles and characteristics, and an experiment of a 2 kW prototype was conducted, the results of which demonstrate the advantages of the presented circuits.

**Keywords:** ZVZCS; PSFB converter; high-power converters; duty cycle loss

## 1. Introduction

The application of electric vehicles has become a reality with the rapid development of lithium battery technology. The battery is a core component in electric vehicle (EV) systems, and questions of how to shorten the charging time and prolong battery life are still bottlenecks in EV systems. As such, developing a charger with high performance over a wide range is a hot issue in power electronic research fields. Nowadays, LLC (resonant converter) and PSFB converters are popular typologies for battery chargers. The LLC converter features high efficiency and no reverse recovery problem of rectifier diodes, which are popular in low- and medium-power applications [1–3]. But the output range is narrow due to the limited switching frequency range and flattened voltage gain, meaning this is not the best choice for the battery charger. PSFB converters, meanwhile, can adjust the output and achieve soft-switching over a wide load range, meaning they may be more suitable for a lithium battery charger. However, PSFB converters still have drawbacks that cannot be ignored, such as the narrow soft-switching range of the lagging-leg switches and a high duty cycle loss. Recently, many studies have focused on these topics, seeking to enhance the performance of PSFB converters [4–8].

The circuits in [9,10] utilize high primary inductance to enlarge the load range of lagging-leg switches in order to realize zero-voltage switching (ZVS). In [9], a coupled inductor with a three-winding transformer provides more resonance energy and ensures the realization of ZVS in lagging-leg switches. It can also decrease the voltage spikes of the rectifier diodes. Reference [10] proposes a new rectifier structure with five diodes, which uses two transformers and two split capacitors on the primary side. This circuit achieves a high voltage gain as well as high efficiency. But the structures in [9,10] are a little complex and may cause extra power loss during the conduction period as well as duty cycle loss. In [11,12], the realization of a better ZVS performance and less duty cycle

loss relies on two diodes conducting the primary current during free-wheeling periods. With the combination of a resonant half-bridge and pulse-width modulation (PWM), the circuit in [13] can eliminate the free-wheeling current and enlarge the soft-switching load range. The auxiliary transformer in [14] decreases the current through clamping diodes and has similar soft-switching characteristics to [12]. Reference [15] utilizes two transformers to help the soft condition of the primary switches, which feature low conduction and duty cycle loss. Two LC networks cross-connect with two input-series–output-parallel PSFB modules, which permit high DC link voltage and guarantee high efficiency over a wide load range [16]. Furthermore, an improved control scheme that has been proposed can reduce the power loss caused by a circulating current by adjusting the phase shift angle [17]. In 2006, reference [18] proposed a new PSFB with a minimum circulating current. Reference [19] focuses on the hard-switching condition and the cycling current. The circuit uses a sharing bridge leg in the primary side and a hybrid rectifier in the secondary side. Moreover, the circuit has different operation modes under different load conditions. It uses a blocking capacitor to limit the cycling current and realizes ZVS of all the switches; thus, efficiency can be improved. There are two transformers in this circuit, and the design requirements of the transformers are relatively high because of the different working modes [20].

Compared with ZVS PSFB converters, zero-voltage and zero-current switching (ZVZCS) PSFB converters can avoid some inherent defects and thus have seen wide application in high-power fields [21–24]. In [23], the ripple voltage provided by a blocking capacitor is used to reset the primary current, and the reverse current is prevented by a two-diode series connecting to the lagging-leg switches. Then, a good soft-switching characteristic over a wide load range can be achieved in the lagging-leg switches. Furthermore, this circuit has minimum duty cycle loss as well as conduction loss. To better reset the primary current during the free-wheeling period, the circuit in [24] adds a clamping diode and a MOSFET behind the rectifier diodes. A controllable voltage source is used in [25] to help quickly reset the primary current, which exceeds the power rating limitation of the conventional ZVZCS converter. In [26], a wide load range of soft-switching is realized in a new topology, combining a half-bridge LLC and PSFB. Furthermore, this converter also features minimum duty cycle loss.

Duty cycle loss has a great impact on power transformation. The center-tapped clamp helps with resetting the primary current and the zero-current switching (ZCS) of the lagging-leg switches. In addition, the circuit can also reduce duty cycle loss and conduction loss [27]. In [28], the saturable inductor is substituted with two diodes; thus, the reverse current of the primary side can be prevented. The power loss caused by conduction and duty cycle loss can also be reduced. The circuit in [29] uses a snubber with capacitors and diodes to ensure the reset of the primary current. It also results in a decrease in both duty cycle loss and circulating loss. Reference [30] uses a variable saturation inductor and a near-ideal transformer. The energy in controllable inductance can ensure a wide soft-switching range and reduce conduction loss, and the small leakage inductor of the transformer can eliminate the duty cycle loss.

Recent studies have improved the applicability of PSFB. However, some problems still exist. For example, high duty cycle loss still occurs under low-input-voltage conditions, and the ratio of the transformer is difficult to optimize. This paper proposes the new topology of a ZVZCS PSFB converter, which inserts a variable voltage source in the primary side to minimize duty cycle loss. Section 2 introduces the operation principles, and an analysis is provided of technical aspects in Section 3. In Section 4, an experiment of a 2 kW prototype is reported, with results that demonstrate the performance of the converter; finally, Section 5 presents our conclusions.

## 2. Circuits and Operation Principles

### 2.1. Circuit Configuration

The proposed circuit, depicted in Figure 1, incorporates IGBT and MOSFET switches. $C_{in}$ represents the input capacitor, with $Q_1$ and $Q_2$ forming the leading legs and $Q_3$ and $Q_4$ constituting the lagging legs. $C_1$ and $C_2$ denote the parasite capacitors associated with $Q_1$ and $Q_2$. The body diodes of $Q_1$–$Q_4$ are represented by $D_1$–$D_4$. The primary current of the main circuit is denoted as $i_p$, and $v_p$ signifies the primary voltage. $T_1$, the main transformer, possesses a turns ratio denoted as $k_{T1}$. The auxiliary circuit comprises $Q_{S1}$~$Q_{S4}$ and $T_2$, where $T_2$, the reset transformer, is characterized by a turns ratio of $k_{T2}$. $Q_{S1}$–$Q_{S4}$ collectively form the auxiliary bridges, and $D_{S1}$–$D_{S4}$ represent their respective body diodes. $C_{S1}$–$C_{S4}$ are the body capacitors. The primary voltage of $T_2$ is labeled as vres', while vres designates the secondary side voltage of the reset transformer. $L_{lk}$ stands for the leakage inductance of $T_1$, and $L_o$ represents the output inductor. To streamline the illustration, only pertinent portions of parasitic capacitances and diodes are featured in the figures. $D_{O1}$ and $D_{O2}$ signify the rectifier diodes.

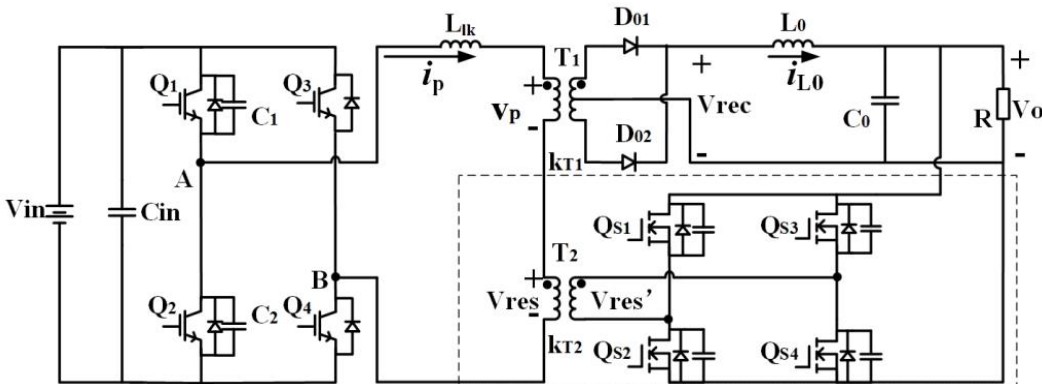

**Figure 1.** Proposed DC/DC converter.

The converter employs the auxiliary circuit to introduce a variable voltage into the primary side. During power transfer stages, this voltage remains zero, with no impact on the operation. However, during free-wheeling stages, this voltage aids in resetting the primary current, ensuring ZCS of the lagging-leg switches and effectively reducing duty cycle losses.

To simplify the analysis, it is assumed that all the components are ideal. $T_1$ and $T_2$ are ideal transformers with a specified turns ratio. The leakage inductance is constant. The magnetizing current of $T_1$ and $T_2$ is low enough to ignore. The parasitic components of $Q_1$, $Q_2$, $Q_3$, and $Q_4$ are of the same value, and $i_{Lo}$ in this circuit can be regarded as a constant current source.

This converter can operate in two different modes according to the switching pattern of $Q_{S1}$ to $Q_{S4}$. These modes are named the normal mode and duty cycle enhanced mode. Figure 2 shows the core waveform of the steady state in the duty cycle enhanced mode, and the waveform of the normal mode is depicted in Figure 3. Figure 4 shows the equivalent circuits during the first half-switching cycle in each mode. Figure 4a–g represent the stages in the normal mode over one half-switching period, and Figure 4a–i give the stages in the duty cycle enhanced mode. In each mode, the operation procedure can be divided into two half periods over one switching cycle, and only one half period is analyzed for simplicity.

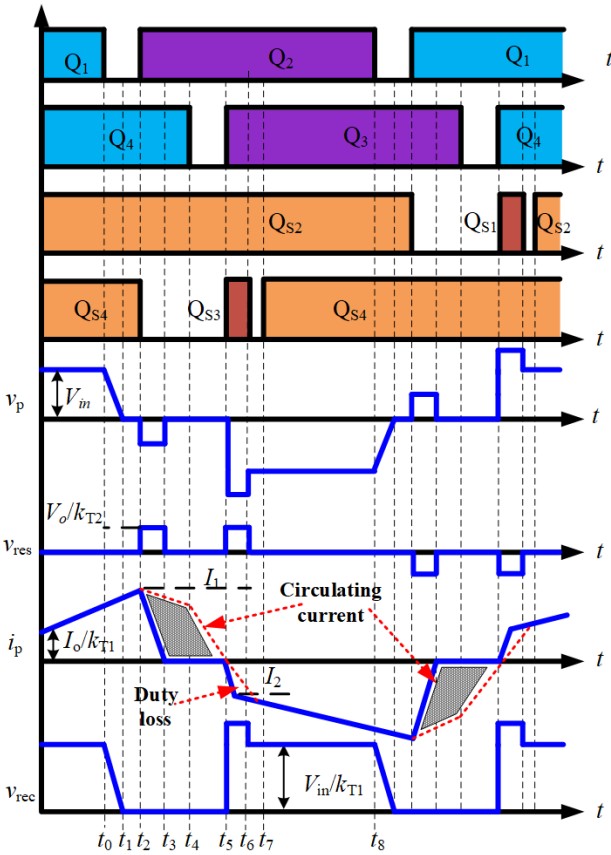

**Figure 2.** Main operating waveform of the proposed circuit in duty cycle enhanced mode.

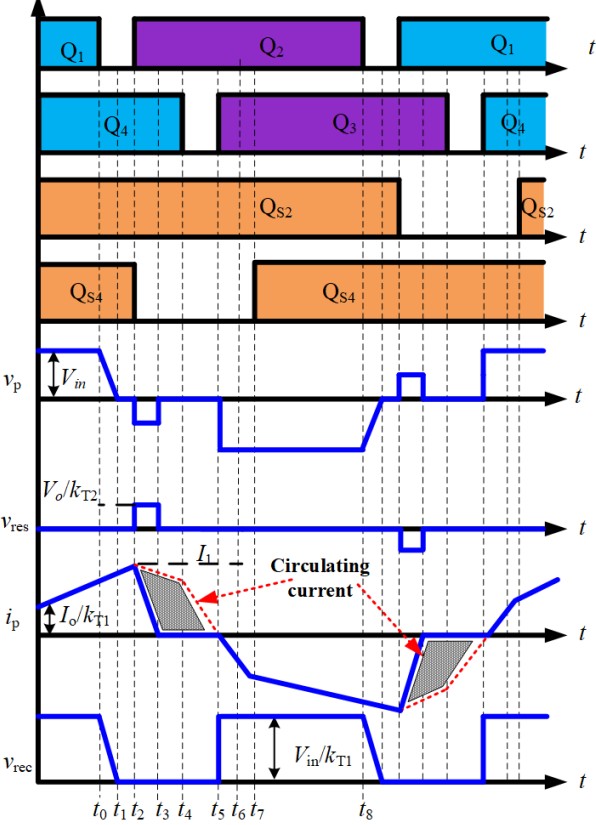

**Figure 3.** Main operating waveform of the proposed circuit in normal mode.

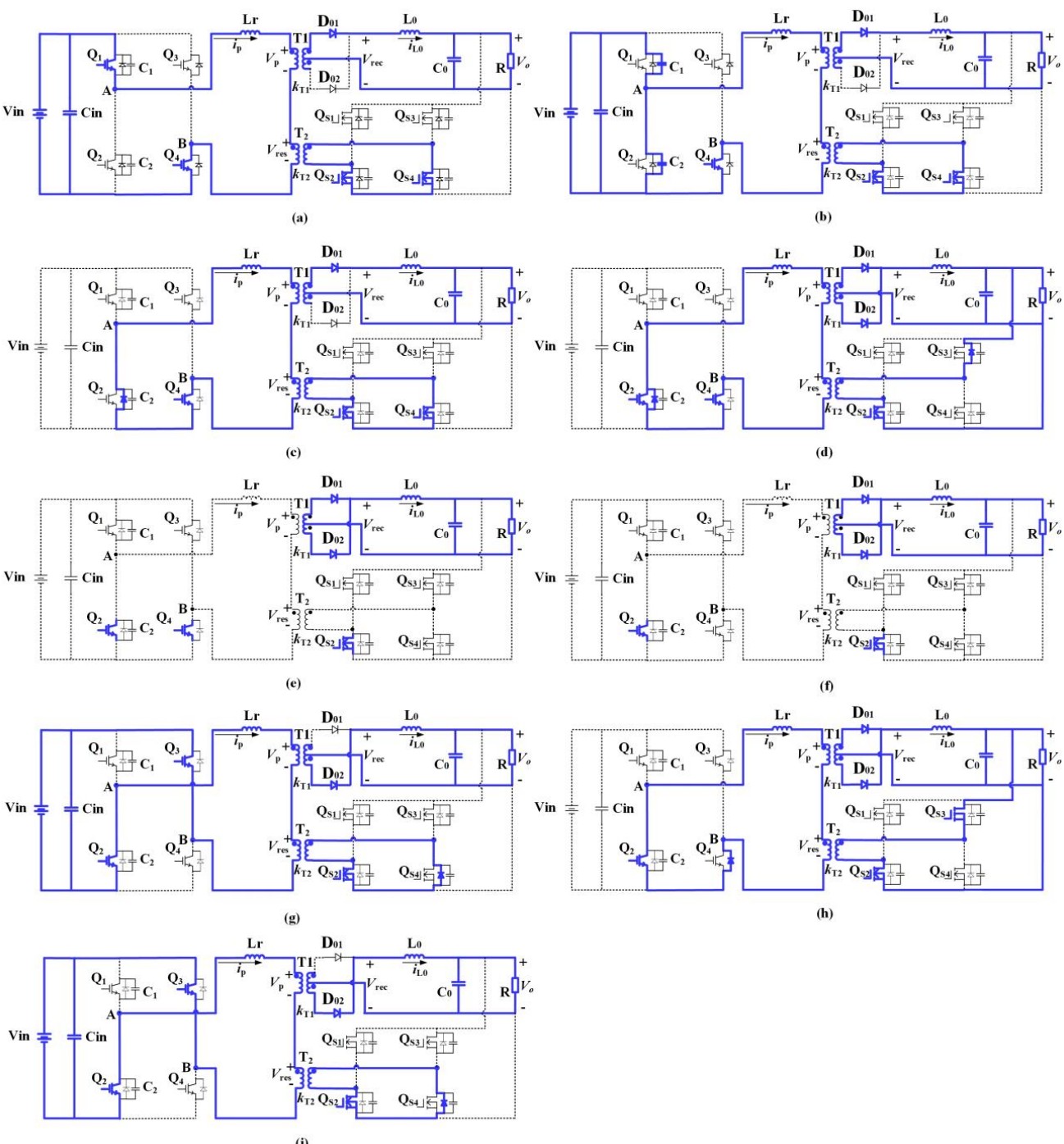

**Figure 4.** Operating circuits in the first half-switching period: (**a**) Stage 1; (**b**) Stage 2; (**c**) Stage 3; (**d**) Stage 4; (**e**) Stage 5; (**f**) Stage 6; (**g**) Stage 7; (**h**) Stage 8 of duty enhanced mode; (**i**) Stage 9 of duty enhanced mode.

### 2.2. Normal Mode

When the circuit works in the normal mode, the primary current can decrease to zero quickly with the help of $v_{res}$, and the ZCS of the lagging-leg switches can be guaranteed.

Stage 1 (Figure 4a, before $t_0$): Before $t_0$, $Q_1$, and $Q_4$ are on, $T_1$ transfers power from the primary side to the secondary side and the load. $Q_{S2}$ and $Q_{S4}$ are also on, the secondary side of $T_2$ free-wheels the primary current, and $v_{res}$ is zero.

In this mode, the circuit works in the normal power transformation period, and $i_p$ is decided by $i_o$. This stage ends when $Q_1$ turns off.

Stage 2: (Figure 4b, $t_0$–$t_1$): At $t_0$, $Q_1$ turns off with ZVS due to $C_1$. Then, $C_1$ charges and $C_2$ discharges during this stage. At the end of this stage, the voltage of $C_2$ returns to zero and the voltage of $C_1$ becomes $V_{in}$. The charge and discharge of $C_1$ and $C_2$ are due to the high value of $L_o$. This mode finishes when the voltage of $Q_2$ is zero. In this mode, $v_p$ decreases with the following rate:

$$v_p = V_{in} - \frac{1}{C_1 + C_2} i_p(t_1 - t_0) \tag{1}$$

Stage 3 (Figure 4c, $t_1$–$t_2$): In this stage, the voltage of $Q_2$ is zero and the circuit is still on with the help of the body diode. $i_p$ flows through $D_2$ and $Q_4$, and the primary circuit turns into the free-wheeling mode. $i_p$ and $i_{T2}$ stay unchanged.

Stage 4 (Figure 4d, $t_2$–$t_3$): $i_p$ is resetting in this stage, and $Q_4$, $D_{S3}$, and $Q_2$ are on. At $t_2$, $Q_2$ is on with ZVS, and $Q_{S4}$ is off simultaneously. Because of $C_{S4}$, the voltage of $Q_{S4}$ cannot change sharply, and the switching-off loss is low. A partial current of $Q_{S4}$ turns to $D_{S3}$.

$$\begin{cases} v_{res} = k_{T_2} V_0 \\ v_p = -v_{res} = -k_{T2} V_0 \\ v_{rec} = 0 \end{cases} \tag{2}$$

During this stage, $v_{res}$ is applied to $L_{lk}$, and $i_p$ decreases linearly. When $i_p$ is lower than $I_o k_{T1}$, $I_{Lo}$ goes through $D_{O1}$ and $D_{O2}$, and the secondary side is shorted.

$$\begin{cases} i_p(t) = i_p(t_2) - \frac{V_0 k_{T2}^2 (t-t_2)}{L_{lk}} = \frac{I_0}{k_{T1}} - \frac{V_0 k_{T2}(t-t_2)}{L_{lk}} \\ i_{T2}(t) = \frac{I_0 k_{T2}}{k_{T1}} - \frac{V_0 k_{T2}^2 (t-t_2)}{L_{lk}} \end{cases} \tag{3}$$

$$\begin{cases} i_{D_{o1}}(t) = I_0 - \frac{V_0 k_{T2} k_{T1}(t-t_2)}{L_{lk}} \\ i_{Do3}(t) = \frac{V_0 k_{T2} k_{T1}(t-t_2)}{L_{lk}} \end{cases} \tag{4}$$

In this circuit, the primary current $i_p$ can reset fast, entering the free-wheeling periods, and as shown in Figure 2, the circulating current can also decrease and is much lower than that of the conventional PSFB converter. In addition, $Q_3$ and $Q_4$ obtain a wide load range of ZVZCS.

Stage 5 (Figure 4e, $t_3$–$t_4$): At $t_3$, when $i_p$ is zero, the current of the auxiliary circuit also decreases to zero. The body diode of $Q_{S3}$ turns off and the reset voltage $v_{res}$ becomes zero.

Stage 6 (Figure 4f, $t_4$–$t_5$): During the last stage, the current of the primary side is zero. $Q_4$ turns off with ZCS at $t_4$.

Stage 7 (Figure 4g, $t_5$–$t_7$): Because $L_{lk}$ limits the varying rating of $i_p$, $Q_3$ realizes a quasi-ZCS turning on at $t_7$.

$$i_p(t) = -\frac{V_{in}}{L_{lk}}(t - t_6) \tag{5}$$

### 2.3. Duty Cycle Enhanced Mode

Stages 1–7 in this mode are the same as in the normal mode.

Stage 8 (Figure 4h, $t_5$–$t_6$): $Q_{S3}$ turns on at $t_7$ and the current of the primary side $i_p$ flows through $Q_2$ and $Q_4$ and starts to increase in the opposite direction. The reset voltage overlays with $v_p$, thus accelerating the reverse growth of $i_p$, and the transformation of power can restart quickly. Because of the reduction in time of the freewheeling of the secondary side, the duty cycle loss can be largely reduced.

$$\begin{cases} v_{res} = V_0 k_{T2} \\ i_p(t) = -\frac{v_{res} + v_p}{L_r}(t - t_5) \\ T_{65} = L_{lk} I_2 / v_{res} \end{cases} \tag{6}$$

Stage 9 (Figure 4i, $t_6$–$t_7$): $Q_3$ is on at $t_6$, and $Q_{S3}$ is off. The primary side transfers power to the load, and $D_{o1}$ is off because of the reverse voltage. Current in the auxiliary circuit flows through $Q_{S3}$ and $D_{S4}$, and $v_{res}$ is zero. After $t_7$, the circuit works in the next half stage.

In this mode, the reset voltage working as an excitation source creates a rapid growth of current to reduce the duty loss. In Figure 2, the dotted line is the waveform of the conventional PSFB, which grows much slower than the duty-enhanced working mode.

## 3. Performance Analysis and Comparison

### 3.1. Duty Cycle Loss

Duty cycle loss is the major problem of the conventional PSFB converter. It happens when $i_p$ cannot change immediately owing to $L_{lk}$. Details of the duty cycle compensation in the proposed converter are given in Figure 5. The duty cycle loss is

$$t_7 - t_5 = T_s \cdot \Delta D = \frac{2L_{lk} \cdot I_o}{n_1(V_{in} + V_{res})} \tag{7}$$

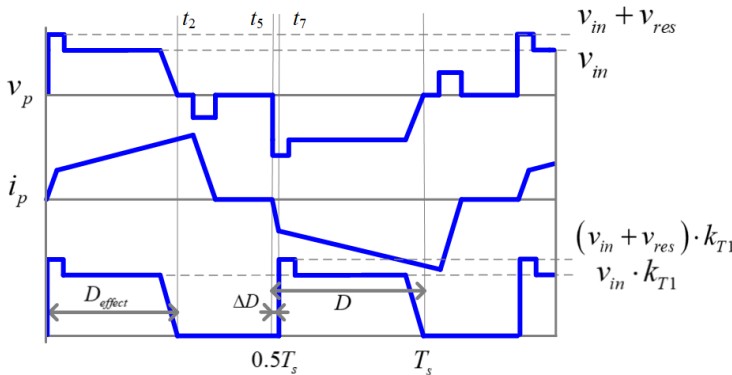

**Figure 5.** Calculation of duty cycle loss.

The actual duty cycle is

$$D_{eff} = D - \Delta D \tag{8}$$

In Equation (8), $D$ is the expected value of the duty cycle and $D_{eff}$ is the actual value. The output voltage is

$$V_o = \frac{D_{eff} \cdot V_{in}}{n_1} \tag{9}$$

### 3.2. Condition of Soft-Switching

#### 3.2.1. ZVZCS of $Q_1$ to $Q_4$

The leading-leg switches obtain ZVS easily with a high value of equivalent inductance. Hence, $Q_1$ and $Q_2$ ensure ZVS turning on over a wide load range. The condition of ZVS is

$$\begin{cases} \frac{1}{2}L_p\left(\frac{I_o}{k_T}\right)^2 \geq \frac{1}{2}(C_1 + C_2)V_{in}^2 \\ L_p = L_{lk} + n_1^2 L_o \end{cases} \tag{10}$$

The minimum load current to ensure ZVS is

$$I_{o,\min} = k_T V_{in}\sqrt{\frac{C_1 + C_2}{\left(L_{lk} + k_T^2 L_o\right)}} \tag{11}$$

When $Q_1$ and $Q_2$ are turned off, $C_1$ and $C_2$ will restrain the rising speed of the voltage. Therefore, these switches operate with quasi ZVS turn-off.

$Q_3$ and $Q_4$ can achieve ZCS because $i_p$ decreases to zero before they are off. The rising speed of $i_p$ is restrained by $L_{lk}$ and $v_{res}$. The condition of ZCS is

$$T_{phase} - T_{dead} \geq \frac{L_{lk}I_2}{V_{res}} \tag{12}$$

$v_{res}$ appears only in the free-wheeling modes, and it does not place much electrical stress on the main components [17,18].

When $Q_3$ and $Q_4$ are on, these switches can realize quasi-ZCS since $i_p$ cannot change immediately.

### 3.2.2. ZVZCS of Auxiliary Switches

The operation principles of $Q_{S1}$ and $Q_{S3}$ are the same, and the situations of $Q_{S2}$ and $Q_{S4}$ are identical.

In the normal mode, $Q_{S1}$ and $Q_{S3}$ are off permanently. $Q_{S2}$ and $Q_{S4}$ turn off with quasi-ZVS and turn on with ZCS. Hence, the switching loss is very low. In the duty cycle enhanced mode, $Q_{S2}$ and $Q_{S4}$ turn off and on with quasi-ZVS. $Q_{S1}$ and $Q_{S3}$ are on with quasi-ZCS and off with quasi-ZVS. Therefore, the power of $Q_{S1}$ to $Q_{S4}$ is a little higher than that of the normal mode.

### 3.3. Comparison

#### 3.3.1. Consideration

This paper uses the classical topology of the ZVZCS PSFB converter and ZVS PSFB converter, shown in Figure 6, to evaluate the proposed converter. The ZVZCS PSFB and the ZVS PSFB chosen here are common PSFBs (refer to [22,29,30]).

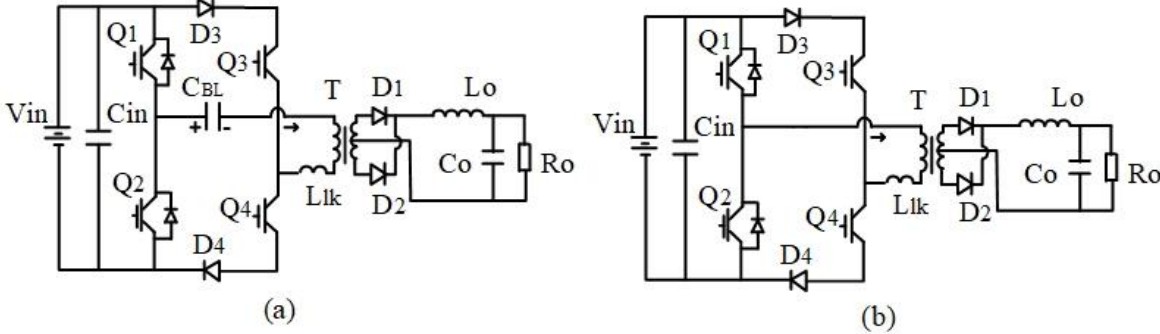

**Figure 6.** Conventional converter for comparison: (**a**) conventional ZVS converter; (**b**) conventional ZVZCS converter.

The comparison is made with these baselines: the input voltage is 500 V (±20%), $V_o$ is 28 V, and $P_o$ is 10 or 100 kW. The value of $L_{lk}$ is 10 μH, and the duty cycle is 1.

#### 3.3.2. Performance Comparison

For the circuit in Figure 6a, the voltage press of the rectifier diodes will be influenced by the reset voltage of the primary current. During the reset course, the reset voltage will be added to the voltage press of the diodes. However, in the proposed circuit, the reset voltage will only appear in the free-wheeling stage and will not influence the rectifier diodes. For the circuit in Figure 6b, the components of the primary side will suffer from an inrush current in operation. Both the reset voltage and the inrush current will increase the power loss of the circuit, and the dash current will apparently limit the power range of the operation condition. As a result, for these two compared circuits, the soft-switching ranges are narrower than for the proposed circuit because of the loss of duty cycle and the limitation of the power rating. The detailed calculation of the loss of duty cycle below shows that the smallest loss is for the proposed circuit.

As shown in Table 1, the number of components in the proposed circuit is larger than those for the circuits in Figure 6a,b, which is the main drawback of this converter. However, the proposed converter can reset the primary current under any power rating with a specific design of $T_2$, which may be the only choice in high-power applications. Therefore, it is challenging to apply the proposed converter in high-power industrial applications.

**Table 1.** Performance comparison.

| Item | Figure 6a | Figure 6b | Proposed Circuit |
|---|---|---|---|
| Number of components | 11 | 12 | 13 |
| Conditions of soft-switching | Narrow range | Power rating limitation | Wide load range |
| Duty cycle loss (40 kW) | 0.3 | 0.12 | 0.08 |
| Conduction loss (40 kW) | 80 W | 60 W | 58 W |

As can be seen in Table 1 and Figure 7, the proposed converter has the minimum duty cycle loss. For comparison, the duty cycle losses and turns ratios of other converters were calculated using Equations (13) and (14). When $P_o$ reaches 50 kW, the duty cycle loss of the ZVS circuit is over 0.5 and the primary side will not have power transmission to the secondary side. When $P_o$ is higher than 40 kW, the ZVS converter cannot work normally and the duty cycle loss is beyond the reasonable range. When $P_o$ is higher than 90 kW, the conventional ZVZCS converters have the same situation.

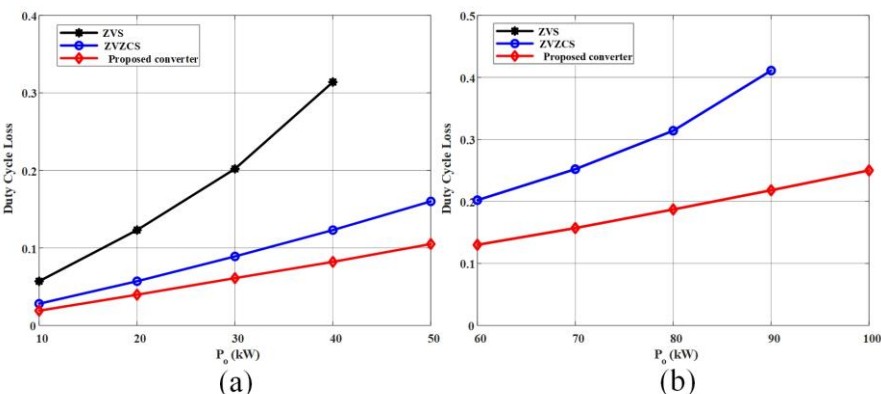

**Figure 7.** Comparison of duty cycle loss: (**a**) duty cycle loss of ZVS, ZVZCS, and proposed converter ($P_o$ = 10 kW–50 kW); (**b**) duty cycle loss of ZVZCS and proposed converter ($P_o$ = 60 kW–100 kW).

The transformer voltage ratio is the key parameter. When the turns ratio is high, in the same condition, the current of the primary side is low and so is the voltage press of the rectifier diodes. Thus, the conduction loss and cost will be low. But a high turns ratio will cause a low output voltage because of the duty cycle loss.

For the converter in Figure 6, $L_{lk}$ causes serious duty cycle loss under high-power and low-input-voltage conditions. Therefore, the turns ratio must be lowered to compensate for the duty cycle loss. However, in the proposed converter, the duty cycle loss can be effectively reduced with the help of the reset voltage $v_{res}$ provided by the auxiliary transformer. Thus, the turns ratio of $T_1$ in the proposed converter can be optimized, and the expected performance of the proposed converter is high.

When the DC bus voltage is 400 V, the duty cycle losses of the three circuits are

$$\begin{cases} \Delta D_{pro} = \dfrac{2L_{lk} \cdot i_p \cdot f_s}{V_{in(\min)} + v_{res}} \\ \Delta D_{ZVZCS} = \dfrac{2L_{lk} \cdot i_p \cdot f_s}{V_{in(\min)}} \\ \Delta D_{ZVS} = \dfrac{4L_{lk} \cdot i_p \cdot f_s}{V_{in(\min)}} \end{cases} \tag{13}$$

$\Delta D_{pro}$ is the duty cycle loss of the proposed converter, $\Delta D_{ZVZCS}$ is the duty cycle loss of the converter in Figure 6a, and $\Delta D_{ZVS}$ is the duty loss in Figure 6b.

The secondary voltage can be calculated as

$$\begin{cases} V_{\text{sec(min)}} = \frac{V_{o\max}+V_D+V_{Lo}}{D_{eff}} \\ n = \frac{V_{in(\min)}}{V_{\text{sec(min)}}} \end{cases} \tag{14}$$

$V_D$ = 1.5 V and $V_{Lo}$ = 0.5 V are the voltage drops on the rectifier inductor. Furthemore, the turns ratios under 10 kW are

$$\begin{cases} n_{pro} = 13.05 \\ n_{ZVZCS} = 12.92 \\ n_{ZVS} = 12.54 \end{cases} \tag{15}$$

where $n_{pro}$ is the turns ratio of $T_1$ in the proposed circuit, $n_{ZVZCS}$ is the turns ratio of the conventional ZVZCS PSFB converter in Figure 6a, and $n_{ZVS}$ is the turns ratio of the conventional ZVS PSFB converter in Figure 6b.

Under 90 kW, the turns ratios are

$$\begin{cases} n_{pro} = 10.7 \\ n_{ZVZCS} = 7.8 \end{cases} \tag{16}$$

As shown in Figure 7, when $P_o$ is under 50 kW, it is obvious that the ZVS converter loses more duty cycle than the conventional ZVZCS converter and the proposed converter. When $P_o$ is higher than 50 kW, the proposed converter has more advantages in duty cycle loss. As shown in Figure 8, this has a great influence on the optimization of the transformer. Compared with the conventional ZVZCS PSFB converter, the optimization of the turns ratio can reach 40%. Moreover, the decrease in the power loss of the primary side can be seen in Figure 9.

To realize a rapid reset of the primary current, this circuit uses a variable voltage provided by the auxiliary circuit. The rapid and effective recovery of $i_p$ can break the power range limitation of the existing ZVZCS converters. Therefore, the proposed converter is well-suited to high-power applications.

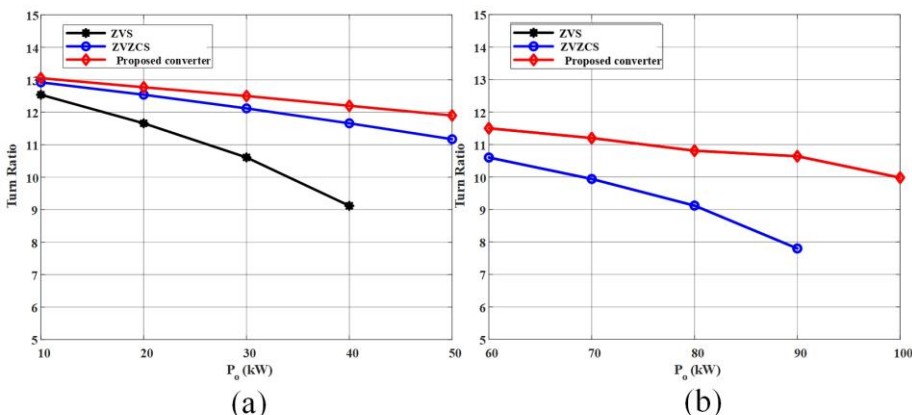

**Figure 8.** Variable turns ratio: (**a**) ZVS, ZVZCS, and the proposed converter ($P_o$ = 10 kW–50 kW); (**b**) ZVZCS and the proposed converter ($P_o$ = 60 kW–100 kW).

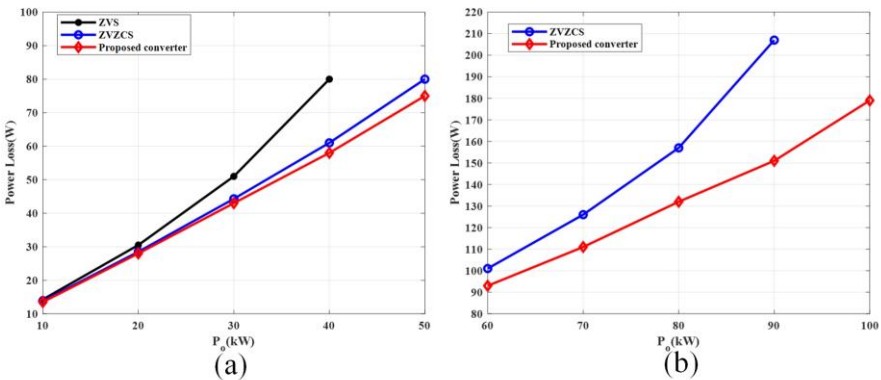

**Figure 9.** Comparison of conduction loss: (**a**) ZVS, ZVZCS, and the proposed converter ($P_o$ = 10 kW–50 kW); (**b**) ZVZCS and the proposed converter ($P_o$ = 60 kW–100 kW).

## 4. Experiments' Results

To verify the proposed converter, a laboratory prototype was built. The key parameters of the circuit are shown in Table 2. The switching frequency is 20 kHz considering IGBT with a high current rating.

**Table 2.** Main parameters of the experiment.

| Item | Parameter |
| --- | --- |
| Rated power | 2 kW |
| Input | 400–600 V |
| Rated output | 28 V/72 A |
| Switching frequency | 20 kHz |
| IGBTs | FF450R12KT4 |
| | MMG150J120UZ6TN |
| kT1 | 10:1 |
| kT2 | 1:6 |
| Magnetic material | Ferrite |
| Volume of T1 | 506 cm$^3$ |
| Volume of T2 | 356 cm$^3$ |
| Turns of T1 | Primary: 20 Secondary: 2 |
| Turns of T2 | Primary: 13 Secondary: 78 |
| QS1, QS2, QS3, QS4 | IXFN110N60P3 |
| DO1, DO2 | MCK400TS60S |
| LO | 10 µH |
| CO | 2000 µF |

Figure 10 shows the experiments' results for the proposed converter.

For the convenience of presentation, the common characteristics are illustrated with the results from the duty cycle enhanced mode, and the special waveform of the normal mode is provided in Figure 10h. As shown in Figure 10a, $v_{res}$ is 168 V after $t_1$, and $i_p$ decreases. The circle with dashed lines is the reset time in Figure 10a. With the turns ratio of T2 changing, the circuit can have different reset voltages and can operate in good conditions when the load changes even in high-power applications, which means no existing power rating limitation. In Figure 10a, $v_p$ has a spike at the beginning of the power transfer stages, which can minimize the duty cycle loss. As the duty cycle enhanced mode is only used in the low-input-voltage mode, this spike does not increase the voltage rating of the rectifier diode. As shown in Figure 10h, there is no voltage spike in the normal mode.

The ZVS operation of $Q_1$ and $Q_2$ is depicted in Figure 10b, and they can realize ZVS over a wide load range with the help of the output inductor. In this picture, $D_1$ conducts and $v_{Q1(CE)}$ reduces to zero before $t_1$; then, $v_{Q1(GE)}$ reaches the threshold voltage at $t_2$. Hence, $Q_1$ and $Q_2$ in the proposed circuit turn on with ZVS.

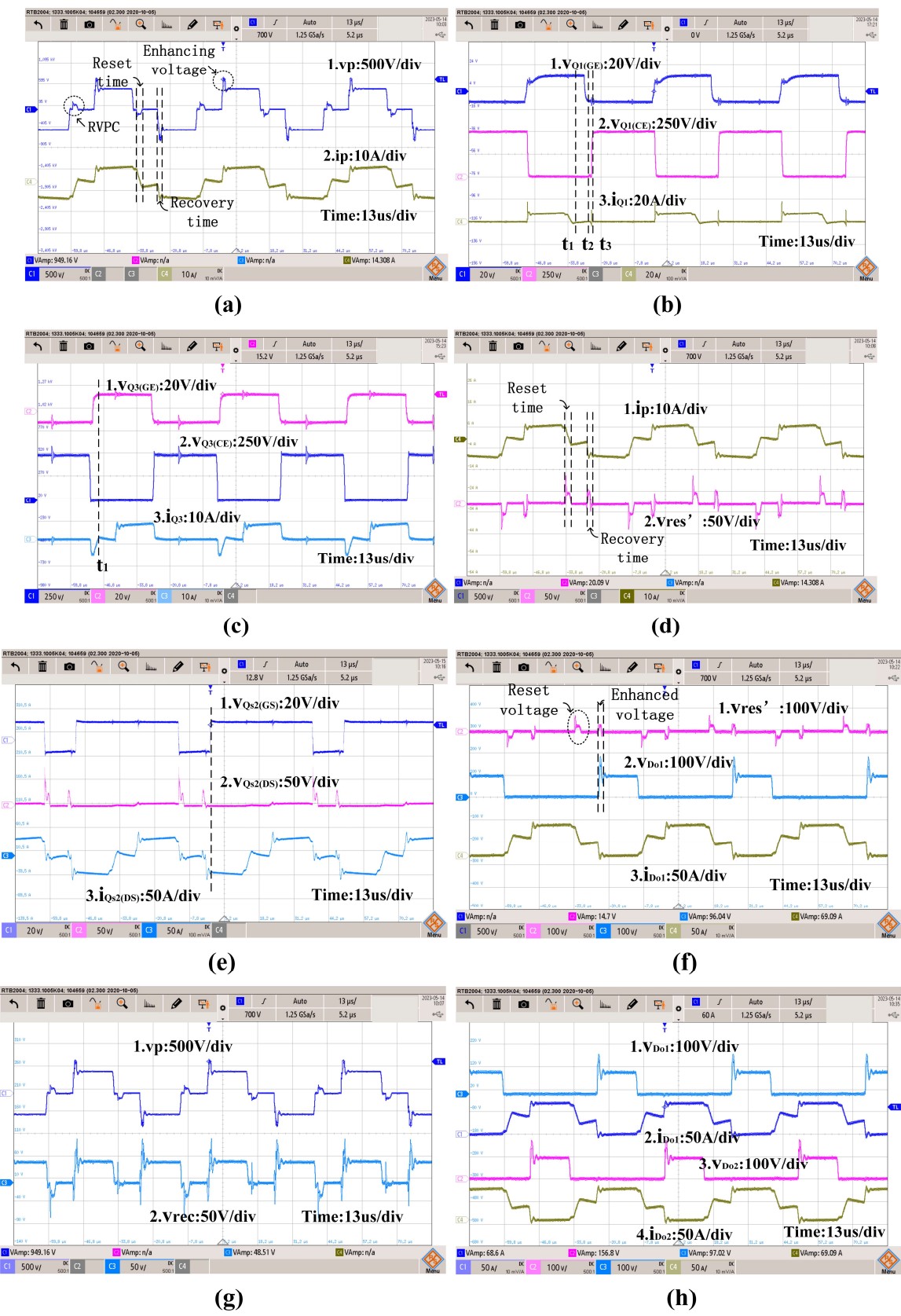

**Figure 10.** Experiments' results: (**a**) $v_\text{P}$ and $i_\text{P}$; (**b**) $v_{\text{Q1(GE)}}$, $v_{\text{Q1(CE)}}$, and $i_{\text{Q1}}$; (**c**) $v_{\text{Q3(GE)}}$, $v_{\text{Q3(CE)}}$, and $i_{\text{Q3}}$; (**d**) $i_\text{P}$ and $v_\text{res}'$; (**e**) $v_\text{res}$, $v_{\text{Do1}}$ and $i_{\text{Do1}}$, and $i_{\text{Qs2}}$; (**f**) $v_{\text{Do1}}$, $i_{\text{Do1}}$, $v_{\text{Do2}}$, and $i_{\text{Do2}}$; (**g**) $v_\text{P}$ and $v_{\text{Do1}}$; (**h**) $v_\text{P}$ and $v_{\text{Do1}}$ in normal mode.

Figure 10c shows that $Q_3$ and $Q_4$ can realize ZCS over a wide load range because $i_p$ is decreased to zero at corresponding switching-off instants. At $t_1$, when $v_{Q3(GE)}$ is about 15 V, $i_{Q3}$ reaches zero. At $t_1$, $v_{Q3(GE)}$ is about $-10$ V, meaning that $Q_3$ is already off with ZCS. In Figure 10c, $v_{Q3}$ has a current spike at $t_3$. Figure 4f,g give the equivalent circuits, and the mechanism of the spike is described as follows. In Figure 10f, $v_{Q3} = v_B = v_A = 0$, and $v_{Q3}$ remains at zero in this stage. After $Q_4$ is on, the input voltage charges $C_3$, which causes a current spike. The energy of the spike will be stored in $C_o$ and will be released at the next instant of switching on. Thus, the power loss is low.

The voltage and current of transformer $T_2$ can be seen in Figure 10d. $v_{res}$ only appears in the current reset mode and the duty cycle enhanced mode. The current of the secondary side is $i_p$ and that of the primary side of $T_2$ is $i_p/k_{T2}$. The integration of $T_1$ and $T_2$ may reduce the volume and may be investigated in future work.

In Figure 10e, the increase in the voltage stress of the rectifier diodes will only happen when duty cycle enhanced mode is applied because the voltage of the DC bus is low, and the enhancing voltage is also adjustable so there is no extra voltage stress added to the diodes, which agrees well with the theoretical analysis. This can be seen in the comparison of Figure 10g,h. Since the efficiency is influenced by the VA rating of the rectifier diodes, the proposed converter will have a better performance in high-power applications.

As shown in Figure 11, the efficiency grows with the load. According to the component parameters, we expected the maximum efficiency of the proposed converter to be about 96%. In Figure 11b, the power loss is about 100 W and the maximum efficiency is around 96%. The rectifier diodes on the secondary side generate about 70 W power loss, which represents the majority of the overall power loss. To further increase efficiency, a synchronous rectifier can be adopted. Figure 12 shows a photo of the prototype. To limit the leakage inductance and have a higher efficiency, $T_2$ uses six small transformers and each of them is 1:1.

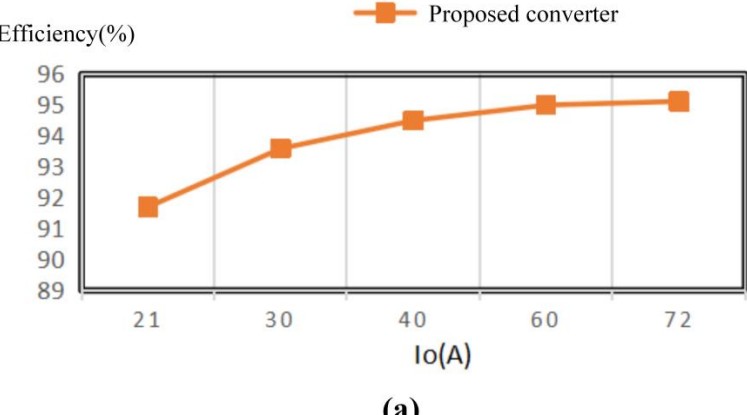

**(a)**

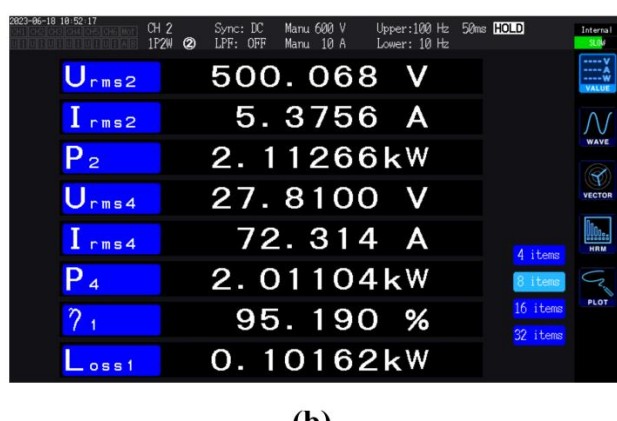

**(b)**

**Figure 11.** Efficiency: (**a**) η with Io; (**b**) hard copy of efficiency.

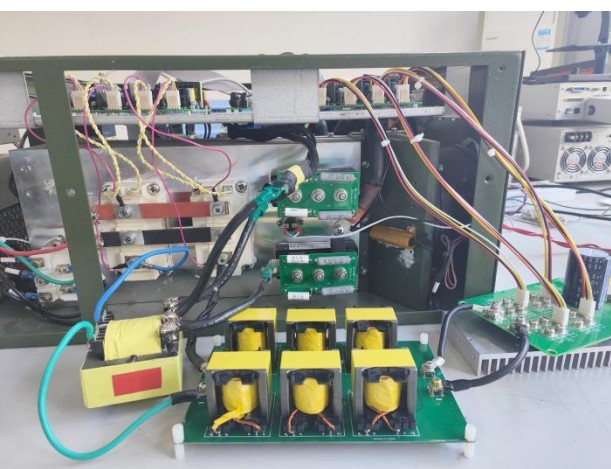

**Figure 12.** Photo of the proposed prototype.

## 5. Conclusions

In this article, we have proposed a ZVZCS PSFB converter, which can reset the primary current and reduce duty cycle loss. For applications in EV batteries, the proposed circuit will minimize duty cycle loss while avoiding significantly increasing the overall power loss of the circuit. Thus, the efficiency of power transmission can be greatly increased and the charging time can be shortened. Furthermore, the realization of ZVZCS can reduce the amount of heat because of power loss and can help prolong the battery life. Specifically, this circuit has the following advantages:

(1) The duty cycle loss can be reduced effectively, and the circuit can be optimized;
(2) The primary switches can realize soft-switching over a wide load range and the additional power loss caused by the auxiliary circuit is low;
(3) The ZVZCS operation has no power rating limitation;
(4) The electrical stress of the components is much lower than that of the conventional ZVZCS PSFB converter.

Further work may include efforts toward the optimization of the auxiliary transformer, lower loss on rectifier diodes, and circuit integration.

**Author Contributions:** Methodology, Y.S.; Resources, Y.S.; Writing—original draft, Y.W.; Writing—review & editing, Y.W., Y.S. and K.X.; Supervision, Y.S. and K.X. All authors have read and agreed to the published version of the manuscript.

**Funding:** This research was funded by the Key Research and Development Projects of Shaanxi Province (grant number 2024GX-YBXM-281).

**Data Availability Statement:** The data presented in this study are available in this article.

**Conflicts of Interest:** The authors declare no conflict of interest.

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
