# Peer review of "A New Zero-Voltage Zero-Current Switching Converter with Minimum Duty Cycle Loss"

_electronics, doi:10.3390/electronics13030518_

Round 1

Reviewer 1 Report

Comments and Suggestions for Authors

1.      In the paper, a new ZVZCS PSFB converter, which uses an auxiliary circuit to minimize the duty cycle loss was proposed. The auxiliary circuit is built of four MOSFETs and an auxiliary transformer, and extra voltage can be added to the primary coils the fast the change rate when the DC voltage is low.

2.    Please compare the contributions of the proposed technology to related technologies, in detail.

3.    In the figure 4, Operating circuits in the first half switching period: a) Stag1; b) Stage 2; c) Stage 3; d) 138 Stage 4; e) Stage 5; f) Stage 6; should be elaborated in detail.

4.      In the table 1, performance comparison should be elaborated in detail.

5.    The manuscript has only 13 pages, the number of the pages should be increased to retain the reader interest. In my view, the appropriate manuscript length is 15 pages. The manuscript should be refined by adding some content.

6.    Please thoroughly revise the language before your submission. 

Comments on the Quality of English Language

Minor editing of English language required.

Author Response

Please see this attachment.

Reviewer 2 Report

Comments and Suggestions for Authors

The paper proposes a new ZVZCS PSFB converter, which uses an 11 auxiliary circuit to minimize the duty cycle loss. The auxiliary circuit is built of four  MOSFETs and an auxiliary transformer.

The paper is well organized and ready for publishing after few explanations:

1. The components - D1 and C1 that are parasitic components of Q1 transistor. Are they added in a real circuit or only do you take into account of them? It is not very clear.

2. In explanations must be added that the circuit from Fig. 1 is composed by Bipolar and MOSFET transistors.

3. I am not agree with the symbol of diode overlapped by a black line from Anode-Cathode - that suggests an Anode - Cathode short-circuit.

4. In Fig. 1 and Fig 4 - the MOSFET transistors - are n type channels? Take care on each symbol.

5. Please make more visible Fig. 4 and Fig. 10.

6. Revise spaces between words in entire paper.

7. Add 2 more recent citing resources and please comment them:

- J. Luo, Z. Guo, W. Zhan and S. Chen, "Efficient Hybrid Dual Full-Bridge DC–DC Converters for Pulsed Output Current Applications," in IEEE Transactions on Industrial Electronics, vol. 70, no. 12, pp. 12254-12266, Dec. 2023, doi: 10.1109/TIE.2023.3234134.

- Lakshmi, P.V.; Musala, S.; Srinivasulu, A.; Ravariu, C. Design of a 0.4 V, 8.43 ENOB, 5.29 nW, 2 kS/s SAR ADC for Implantable Devices. Electronics 202312, 4691. https://doi.org/10.3390/electronics12224691

 I recommend the paper publishing after corrections. 

Reviewer 3 Report

Comments and Suggestions for Authors

This study proposed a new Zero-voltage zero-current switching (ZVZCS) phase-shifted full-bridge (PSFB) converter that minimizes duty cycle loss in ZVZCS PSFB by resetting the primary current. The paper is well-written and outlined. I have some comments for the authors to consider.

1.      I recommend the authors check through the manuscript, fix all formatting issues, and ensure that all acronyms are defined when they are first used. For example, please define MOSFETs in the abstract.

2.      Line 219, what is the basis of the baseline numbers used to compare the ZVZCS PSFB converter and ZVS PSFB converter in Fig.6? I recommend supporting this with a reference if applicable.

3.      Some of the figures in the manuscript are ineligible. Please improve the eligibility of Fig.4 and Fig.10 and fix the Fig.4 caption.

4.      Can the authors explain why the ZVS converter does not exist in Figs. 7b, 8b, and 9b when Po exceeds 50kW?

5.      What is the implication of these findings as related to shortening the charging time and prolonging the battery life of EV systems? The conclusion section should be expanded to comment on and provide insights on this.

Comments on the Quality of English Language

The authors should check through carefully for minor editing.

Round 2

Reviewer 1 Report

Comments and Suggestions for Authors

no further comment.

Reviewer 2 Report

Comments and Suggestions for Authors

Authors replied to my questions. Thank you.